# High Serum Adrenomedullin and Mid-Regional Pro-Atrial Natriuretic Peptide Concentrations in Early Pregnancy Predict the Development of Gestational Hypertension

**DOI:** 10.3390/diagnostics14232670

**Published:** 2024-11-27

**Authors:** Aleksandra Jagodzinska, Agnieszka Wsol, Agata Gondek, Agnieszka Cudnoch-Jedrzejewska

**Affiliations:** 1Chair and Department of Experimental and Clinical Physiology, Laboratory of Centre for Preclinical Research, Medical University of Warsaw, 02-097 Warsaw, Poland; 21st Department of Obstetrics and Gynecology, Medical University of Warsaw, 02-097 Warsaw, Poland; 3Department of Methodology, Laboratory of Centre for Preclinical Research, Medical University of Warsaw, 02-097 Warsaw, Poland

**Keywords:** adrenomedullin, pregnancy, gestational hypertension, natriuretic peptides

## Abstract

Objectives: Adrenomedullin (AM) and natriuretic peptide levels are elevated in pre-eclampsia. The aim of the present study was to determine AM and natriuretic peptide concentrations before 20 weeks of pregnancy in women who later developed gestational hypertension and in normal pregnancies. Methods: 95 pregnant Caucasian women were included in the study. Gestational hypertension (GH) was diagnosed in 18 patients. The control group consisted of 41 patients with normal pregnancies (non-GH). Blood samples were taken during the first trimester of pregnancy. Results: Analysis of NT-proBNP showed no significant differences between the group of patients who later developed GH and those with normal pregnancies. Patients who developed GH later in pregnancy had higher levels of both MR-proANP (*p* < 0.001) and adrenomedullin (*p* < 0.001). Higher levels of MR-proANP were found in the GH with pre-eclampsia group compared with the GH without pre-eclampsia group. Higher levels of AM (*p* < 0.05) and MR-proANP (*p* < 0.005) correlated with the risk of preterm birth. Conclusions: (1) Plasma adrenomedullin and MR-proANP concentrations were higher before the 20th week of pregnancy in women who later developed GH; (2) NT-proBNP concentrations did not differ between women with pregnancy-induced hypertension and normal pregnancies; (3) MR-proANP concentrations were highest in patients who developed pre-eclampsia in advanced pregnancy; and (4) there was a correlation between higher plasma adrenomedullin, MR-proANP concentrations before the 20th week of pregnancy, and the risk of preterm birth.

## 1. Introduction

Gestational hypertension (GH) occurs in 6% to 17% of healthy nulligravid women and in 2% to 4% of multigravid women [1,2,3]. It poses a significant risk to the health and life of both mother and child. According to the World Health Organization [4], high blood pressure and its complications are responsible for 16% of the deaths of pregnant women in the developed world. Low birth weight, preterm birth, placental abruption, eclampsia, intrauterine fetal demise, and prolonged neonatal intensive care are some of the most common complications of high blood pressure [5].

GH is diagnosed when systolic blood pressure ≥ 140 mmHg and/or diastolic blood pressure ≥ 90 mmHg occurs after 20 weeks of pregnancy and persists for less than 12 weeks postpartum [6,7]. Hypertension with a range of symptoms such as proteinuria and organ dysfunction, occurring after 20 weeks of pregnancy, during labor, or postpartum is referred to as pre-eclampsia. For more information on the diagnosis of this pathology, please refer to Gondek et al. [8].

During pregnancy, the spiral arteries perfuse the endometrium through the uterine arteries and play a significant role in controlling blood flow to the uterus and the placenta [9]. The uterine artery’s response to vasoactive substances and changes in vascular tone greatly affect this vascular adaptation. An increase in blood flow to the uterus during pregnancy is linked to a reduction in vascular resistance of the uterine artery. Therefore, proper functioning of the uterine artery is crucial for the progression of a healthy pregnancy, and reduced blood flow in the uterus has been associated with miscarriage and pregnancy problems such as pre-eclampsia. The underlying cause of pre-eclampsia is not fully understood. Incomplete remodeling of the spiral arteries, a frequent pathological observation in the pre-eclamptic uterus, can impede blood flow to the placenta [10,11]. Placental ischemia and oxidative stress result in the release of placental factors into the maternal circulation, triggering inflammation, impaired blood vessel function, endothelial dysfunction, and systemic complications [12,13].

Adrenomedullin (AM) is an endogenous peptide first identified as a potent vasodilator molecule. The main sites of the synthesis and release of AM are the smooth muscle cells and the endothelial cells of the vascular system [14]. AM is involved in a wide range of physiological processes, including vasodilation, angiogenesis, inhibition of apoptosis, and regulation of cell growth. AM also has anti-proliferative effects and protects a variety of cells from stressor-induced oxidative stress, suppressing the cAMP pathway [15,16,17]. Experimental studies have shown that AM is an endothelium-delivered vasorelaxant factor. It plays an important role in reducing vascular contraction to facilitate vascular adaptation in rat pregnancy [18]. Adverse pregnancy outcomes in rats due to impaired AM function, and the association of complicated pregnancies, such as pre-eclampsia and intrauterine growth restriction (IUGR), with reduced levels and function of these peptides, suggest an important role for these peptides in facilitating healthy pregnancies. The production of AM is upregulated in human pregnancy as determined by maternal plasma concentrations [19,20], and placental tissues have been shown to be an important site of production in human pregnancy [21]. It has also been shown that the vascular dysfunction and hypertension that occurs in pre-eclampsia may result in an impairment of the AM receptor system. This is in addition to the reported impaired levels of these peptides in serum and the placenta in pre-eclamptic pregnancies. Impaired function of these potent hypotensive peptides in the uterine vasculature may affect uteroplacental perfusion and lead to placental insufficiency. This is one of the major causes of pre-eclampsia and IUGR [22,23,24].

Natriuretic peptides (NP) are hormones that are primarily involved in regulating the water and electrolyte balance and controlling the circulatory system. To date, six classical NPs have been described: A-type natriuretic peptide (ANP), urodilatin, B-type natriuretic peptide (BNP), C-type natriuretic peptide, D-type natriuretic peptide, and uroguanylin. In clinical practice, the levels of the by-products of pro-peptide hydrolysis are determined, i.e., *N*-terminal pro-B-type natriuretic peptide (NT-proBNP) and mid-regional pro-atrial natriuretic peptide (MR-proANP) fragments. Both NT-proBNP and MR-proANP are biologically inactive molecules with a longer half-life in human serum than BNP and ANP and are, therefore, used to determine NP levels. It is worth noting that NPs are secreted in large quantities by cardiomyocytes following myocardial stretching. This is due to volume and pressure overload activating the endocrine system of the heart [25,26]. Plasma concentrations of both peptides are increased in patients with asymptomatic and symptomatic left ventricular dysfunction. This makes their use in the diagnosis of heart failure possible [27]. Elevated levels of NP are associated with a higher incidence of maternal cardiovascular complications and preterm birth [28]. Furthermore, some authors have reported that high NT-proBNP levels during pregnancy are associated with pre-eclampsia, its clinical severity and complications, including life-threatening maternal complications [29,30]. High BNP levels in pre-eclampsia reflect pathophysiological changes in left ventricular mass and volume in pregnant women [31].

As mentioned, the levels of adrenomedullin and natriuretic peptides are elevated in pre-eclampsia. Unfortunately, information is lacking on the possible relationship between plasma concentrations of these peptides in early pregnancy (less than 20 weeks) and the risk of gestational pregnancy and related pregnancy complications. We therefore wondered whether, at an early stage of maternal-fetal unit and spiral vessel development, AM and NP levels might also be elevated before the onset of symptomatic pregnancy-induced hypertension. In this context, our study aimed to determine AM and NP concentrations before 20 weeks in women with gestational hypertension and normal pregnancies.

Today, most professional organizations recommend screening for pre-eclampsia risk in the first trimester. The American College of Obstetricians and Gynecologists (ACOG) and the National Institute for Health and Care Excellence (NICE) have suggested screening for pre-eclampsia based on maternal risk factors. Chronic hypertension, chronic kidney disease, systemic lupus erythematosus (SLE), antiphospholipid syndrome (APS), pre-existing diabetes mellitus, and a history of hypertensive disorders of pregnancy are considered to be severe risk factors, whereas primigravidity, maternal age > 40 years, BMI at first consultation > 35 kg/m^2^, multiple pregnancies, a positive family history of PE and an interpregnancy interval of more than 10 years are considered to be moderate risk factors [4,32]. If one or at least two moderate risk factors for pre-eclampsia are identified, then a low dose of acetylsalicylic acid should be prescribed [33]. The NICE recommendation achieves detection rates of 41% and 34% for preterm and term pre-eclampsia, respectively, with a false positive rate of 10%. Screening based on the 2013 ACOG recommendation achieves detection rates of 5% and 2% for preterm and term pre-eclampsia, respectively, with a false positive rate of 0.2% [4]. The predictive model proposed by the Fetal Medicine Foundation (FMF) combines maternal factors, measurements of mean arterial pressure, uterine artery pulsatility index, and serum placental growth factor. This model has undergone successful internal and external validation. The FMF test has detection rates of 90% and 75% for predicting early and preterm pre-eclampsia, respectively, with a false positive rate of 10% [34]. All the above tests are only used in the first trimester of pregnancy and overlook potential changes that may occur in subsequent trimesters.

## 2. Materials and Methods

### 2.1. Study Population

The study was conducted at the Department of Obstetrics and Gynecology, Medical University of Warsaw. Approval was obtained from the Bioethics Committee of the Medical University of Warsaw, and 95 pregnant Caucasian women were enrolled in the study, of whom 15 patients did not complete the study (due to changing the pregnancy center) and another 21 were excluded from statistical analysis because of the presence of elevated blood pressure (≥140/90 mmHg) before 20 weeks of pregnancy (*n* = 9), hypothyroidism (*n* = 7), or gestational diabetes (*n* = 4) detected at subsequent visits. Gestational hypertension was diagnosed in 18 patients. The control group consisted of 41 patients with normal pregnancies (non-GH).

Blood samples were collected in the first trimester of pregnancy and were analyzed at the Department of Experimental and Clinical Physiology, Medical University of Warsaw. All patients were provided with written information about the study prior to the start of the study and signed an informed consent form for their participation in the study.

Exclusion criteria for the study included: chronic arterial hypertension, chronic diseases in the mother (liver disease, kidney disease, heart failure, diabetes, endocrinological diseases), genetic defects in the fetus, premature leaking of amniotic fluid, intrauterine infection, and multifetal gestation. Gestational age was determined according to the date of the last menstrual period and then corrected according to an ultrasound performed between 11 and 13 weeks of pregnancy. The control group consisted of healthy women with no medical history, no pregnancy complications, and normal blood pressure values measured during follow-up visits.

### 2.2. Biochemical Measurements

Blood samples from pregnant women were stored at room temperature for at least 60 min to allow clotting and then centrifuged at 10,000 rpm for 5 min. The serum obtained was frozen and stored at −80 degrees Celsius. MR-proANP levels were measured by enzyme-linked immunosorbent assay using a commercially available kit: Human MR-proANP ELISA Kit (B9 Bld, High-tech Medical Devices Park, No. 818 Gaoxin Ave. East Lake High-tech Development Zone, Wuhan, China, number: EH4063); detection range: 156–10,000 pg/mL; the intra-assay CV (coefficient of variation) was 8% and the inter-assay CV was 10%. ADM levels were measured by enzyme-linked immunosorbent assay using a commercially available kit: Human AMD ELISA Kit (B9 Bld, High-tech Medical Devices Park, No. 818 Gaoxin Ave. East Lake High-tech Development Zone, Wuhan, China, number: EH0779); detection range: 15.625–1000 pg/mL; the intra-assay CV was 8% and the inter-assay CV was 10%. NT-proBNP levels were determined by Alab Laboratories Sp. z o.o., Warsaw, Poland, with the ROCHE diagnostic test and Cobas instrument (Roche Diagnostics, IN, USA), using the electrochemiluminescence immunoassay method.

### 2.3. Statistical Analysis

Statistica 13.3 software was utilized for conducting statistical analysis and to calculate the area under the receiver operating characteristic (ROC) curves (AUC) to investigate the predictive accuracy of AM, MR-proANP, and NT-proBNP. Statistical analysis included the display of continuous variables as means with standard error. Differences at *p* < 0.05 were accepted as statistically significant. Normality of the distribution was assessed using the Shapiro–Wilk W test. Parametric tests, such as one-way ANOVA and Student’s *t*-test, were used to analyze parameters that followed a normal distribution. The Kruskal–Wallis ANOVA, multiple comparison of means, and Mann–Whitney U tests were used for non-normally distributed parameters.

## 3. Results

### 3.1. Characteristic of the Study Group

Table 1 shows the characteristics of the women in the control (non-GH) and GH groups and their newborns. Compared with women with uncomplicated pregnancies, women in the GH group were older and had a significantly higher body mass index (BMI).

Neonatal weight was lower in the hypertensive group (GH), but when measured in gestational age percentiles, there were no statistically significant differences between GH and the controls (non-GH). In the GH group, 20% of newborns were born with Apgar scores of 8 or less. Such low Apgar scores were not observed in the control group.

In the control group (non-GH), the majority of patients delivered naturally, whereas in the GH group, 70% of pregnancies were terminated by Cesarean section.

All hypertensive patients used methyldopa (in doses ranging from 750 mg to 1500 mg per day), 50% as monotherapy. In total, 25% of the women used a combination of methyldopa (doses as above) and metoprolol (doses of 50 mg to 100 mg per day). In total, 20% of the women used methyldopa and labetalol (at a dose of 200 mg per day), and 5% of the women used methyldopa, labetalol, and nitrendipine (at a dose of 10 mg to 20 mg per day). In 55% of the women, normal blood pressure control was not achieved despite antihypertensive treatment. These patients had their pregnancies terminated early, usually by Cesarean section. Pre-eclampsia was diagnosed in nine patients.

### 3.2. Biochemical Measurements Results

Analysis of the NT-proBNP results before 20 weeks of pregnancy showed no significant differences between the group of patients who later developed gestational hypertension (GH, 80 ± 11 pg/mL) and those with normal pregnancies (non-GH, 55 ± 6.77 pg/mL) (Figure 1A,D).

In the sera of patients who developed gestational hypertension later in pregnancy, higher concentrations of both MR-proANP (GH, 880 ± 110 pg/mL vs. non-GH, 390 ± 30 pg/mL; *p* < 0.001) (Figure 1B,D) and adrenomedullin (GH, 295 ± 74 pg/mL vs. non-GH, 100 ± 4.5 pg/mL; *p* < 0.001) were found (Figure 1C,D).

Multivariate analysis (ANOVA) performed for the control, gestational hypertension without pre-eclampsia, and gestational hypertension with pre-eclampsia groups showed a significant interaction between groups for MR-proANP [H(2, 59) = 17.267; *p* < 0.001] and adrenomedullin [H(2, 59) = 25.910; *p* < 0.001] concentrations. ANOVA failed to show that NT-proBNP levels differed between the abovementioned groups (Figure 2A).

In addition, higher levels of MR-proANP were found in the group of GH with pre-eclampsia (1030 ± 180 pg/mL) compared with the group of GH without pre-eclampsia (740 ± 130 pg/mL; *p* < 0.01) (Figure 2B). AM concentrations did not differ between the two groups (Figure 2C).

In both groups, higher levels of AM and MR-proANP correlated with the risk of preterm birth (AM, *n* = 59, r_s_ = −0.340, *p* < 0.05; MR-proANP, *n* = 59, r_s_ = −0.279, *p* < 0.005). None of the levels of the peptides studied correlated with the BMI of the pregnant women.

## 4. Discussion

This study shows for the first time that (1) plasma adrenomedullin and MR-proANP concentrations were higher before the 20th week of pregnancy in women who later developed gestational hypertension, (2) NT-proBNP concentrations did not differ between women with pregnancy-induced hypertension and normal pregnancies, (3) MR-proANP concentrations were highest in patients who developed pre-eclampsia in advanced pregnancy, and (4) there was a correlation between higher plasma adrenomedullin, MR-proANP concentrations before the 20th week of pregnancy, and the risk of preterm birth.

Vascular remodeling of the uterine spiral arteries during pregnancy is a critical maternal response to ensure adequate blood flow to the growing fetus. The initial stages of the formation of the placenta in humans and rodents occur during implantation when cells from the outer layer of the blastocyst attach to and penetrate the endometrium and transform into extravillous cytotrophoblasts. The extravillous cells invade the endometrium to establish a vascular connection between the fetal placental tissue and the maternal blood supply [35]. High levels of AM are found in trophectoderm cells and persist in trophoblast giant cells in mice, whereas AM expression is observed in the extravillous cytotrophoblast lineage in the human placenta at term [21,36,37,38,39]. Ex vivo studies have shown that AM promotes the growth and movement of a first trimester cytotrophoblast cell line [40]. Furthermore, AM infusion causes vasodilation in a dose-dependent manner, suggesting that AM may help to maintain the low vascular resistance of the placenta [41,42]. Ross et al. found that AM treatment in rats caused relaxation of the uterine artery, with this effect being more pronounced during pregnancy or with estradiol treatment, further supporting the role of AM in regulating vascular tone during pregnancy [43]. We believe that elevated AM levels in early pregnancy in women who later developed gestational hypertension may be related to an increase in adrenomedullin production to counterbalance the increased production and release of vasoconstrictors such as thromboxane A2 and ET-1 from the damaged endothelium. Elevated levels of adrenomedullin may be necessary to regulate placental vascular resistance and/or fetal circulation within normal ranges [20].

Results from human studies are highly variable. Al-Ghafra et al. attempted to clarify the role of AM in pre-eclampsia by limiting their study to patients with severe pre-eclampsia and separating patients by term versus preterm delivery [44]. They found that AM protein levels were increased in fetal membranes in both term and preterm pre-eclamptic patients, and AM mRNA levels were also increased in preterm choriodecidual tissue in pre-eclampsia [44]. A few other studies support our findings. Namely, Senna et al. reported that AM production starts very early in pregnancy, suggesting that it may play an important role in human reproduction from implantation to delivery, and maternal plasma AM levels appear to be higher in pre-eclampsia than in normal pregnancy [45]. However, in contrast to our study, the author did not assess adrenomedullin levels before 20 weeks of pregnancy in a population of women who had developed hypertension during pregnancy. Higher AM levels in hypertensive pregnant women in the third trimester have also been reported by other authors [46,47]. Some data suggest that adrenomedullin synthesis in villous syncytiotrophoblasts is reduced in pregnancies with pre-eclampsia [48]. The reduction in AM mRNAs, in contrast to unchanged mRNA levels of their receptors, has been reported in placental samples from women with pre-eclampsia [49]. Although there is strong evidence from our results, as well as from data from other studies, suggesting that changes in AM levels are either a cause or a secondary effect of pre-eclampsia, it is clear that more controlled experiments are needed to determine the direction of the change and the exact role of AM in pre-eclampsia.

The mechanism for the increase in plasma MR-proANP in hypertensive disorders of pregnancy is unknown. The function of uterine A-type natriuretic peptide is to promote trophoblast invasion and remodeling of the uterine spiral arteries and to maintain normal blood pressure during pregnancy [50]. Another function of A-type natriuretic peptide is to accelerate angiogenesis. It may also be involved in the remodeling of the uterine spiral artery, as it has a protective effect against oxidant-induced injury [51,52,53]. In addition, ANP-deficient pregnant mice have delayed trophoblast invasion, increased blood pressure, and proteinuria characteristic of pre-eclampsia [50]. The source of MR-proANP in pre-eclampsia remains unclear. A study by Degrelle et al. has suggested that the increased circulating A-type natriuretic peptide in women with pre-eclampsia is of placental origin [54]. However, according to Sugulle et al., the increased circulating MR-proANP is of cardiac origin [55]. Their finding of placental A-type natriuretic peptide gene expression in patients and in a rat model convincingly demonstrated that the uteroplacental unit is not the main source of circulating MR-proANP in pregnancy. There are studies showing that MR-proANP is significantly increased in maternal plasma in pre-eclampsia compared with normotensive pregnancies. Most of the studies were performed after 24 weeks of pregnancy [55,56,57,58]. Birdir et al. studied maternal serum concentration of MR-proANP at 11–13 weeks of pregnancy. In the pre-eclampsia group, the maternal serum concentration of MR-proANP was not significantly different from the controls. This is in contrast with our results [59].

Human research has suggested that AM protein levels may be elevated in individuals who experience preterm birth. The Di Iorio group has published several research reports indicating higher levels of AM in the amniotic fluid in individuals with preterm premature rupture of membranes (PPROM) and elevated levels of AM in the amniotic fluid in individuals with preterm labor [60,61,62]. Elevated levels of AM in the amniotic fluid during the second trimester have also been observed in patients who went on to deliver preterm. However, a 2009 study by Iavazzo et al. concluded that there was no difference in AM levels between spontaneous preterm birth and PPROM [63]. Unfortunately, no study has yet been published that has investigated the association between adrenomedullin levels in patients with pre-eclampsia and preterm birth. According to the results of our study, there is an association between higher plasma adrenomedullin concentrations before 20 weeks of pregnancy and the risk of preterm birth. Preterm birth was medically indicated as a complication of pre-eclampsia in all cases in our study.

Wellmann et al. analyzed 25 pregnant patients with pre-eclampsia and 120 healthy pregnant women. The results showed significantly higher levels of MR-proANP in women with pre-eclampsia. In addition, the group with pre-eclampsia delivered statistically earlier (34 weeks of pregnancy) than healthy patients (38 weeks of pregnancy) [64]. The same conclusion was reached by Yi et al. From a total of 77 pre-eclampsia patients, 35 hypertensive patients delivered preterm. Plasma MR-proANP was higher in preterm patients compared with full-term patients [65]. We assume that elevated levels of AM and MR-proANP are associated with placental dysfunction leading to hypertension and pre-eclampsia. According to current recommendations, such pregnancies should be closely monitored. In the event of deterioration in maternal or fetal condition, preterm delivery should be considered to reduce maternal and fetal morbidity and mortality.

Recent studies in healthy pregnant women have shown that mean N-terminal pro-B-type natriuretic peptide (NT-proBNP) levels are similar in the first and second trimesters, but significantly lower in the third trimester [66]. In our study, there was no difference in NT-proBNP levels between women with pregnancy-induced hypertension and women with normal pregnancies. Moungmaithong et al. found results similar to ours. They analyzed NT-proBNP concentrations in 1792 pregnancies. 112 of them had pre-eclampsia in the first trimester. There were no significant differences between NT-proBNP concentrations in pre-eclampsia cases and unaffected cases [67]. However, in a cohort study of 4103 women, Hauspurg et al. concluded that higher NT-proBNP levels were associated with a lower risk of gestational hypertension [68]. We believe that the concentration of NT-proBNP increases only in the later stages of pregnancy complicated by pre-eclampsia, and that there is a link between the placental–maternal failure, the flow disturbance itself, and the development of circulatory overload in pre=eclamptic pregnancy.

## 5. Conclusions

MR-proANP and AM are potentially interesting biomarkers in pre-eclampsia, as they are likely to reflect the hemodynamic and cardiovascular changes and endothelial dysfunction that occur in pre-eclampsia. Both of these peptides may be complementary to well-established biomarkers of pre-eclampsia that are predominantly of placental origin, such as sFlt-1 (soluble fms-like tyrosine kinase-1) or PlGF (placental growth factor). However, more extensive studies in larger groups are needed. Also, systematic assessment of these concentrations in each trimester is needed to determine their precise diagnostic value.

## 6. Limitations of the Study

This study has some limitations. It was designed as a prospective study. The number of people who completed the study dropped to 59 due to comorbidities or because they changed the pregnancy center. In relation to the population included in the study, pregnancy-induced hypertension occurred in almost 18% of the study population. However, taking into account that some patients previously had hypertension in an earlier pregnancy and had chosen medical care at an academic center for the current pregnancy, this de-escalation of pregnancies with pregnancy-induced hypertension seems justified.

## Figures and Tables

**Figure 1 diagnostics-14-02670-f001:**
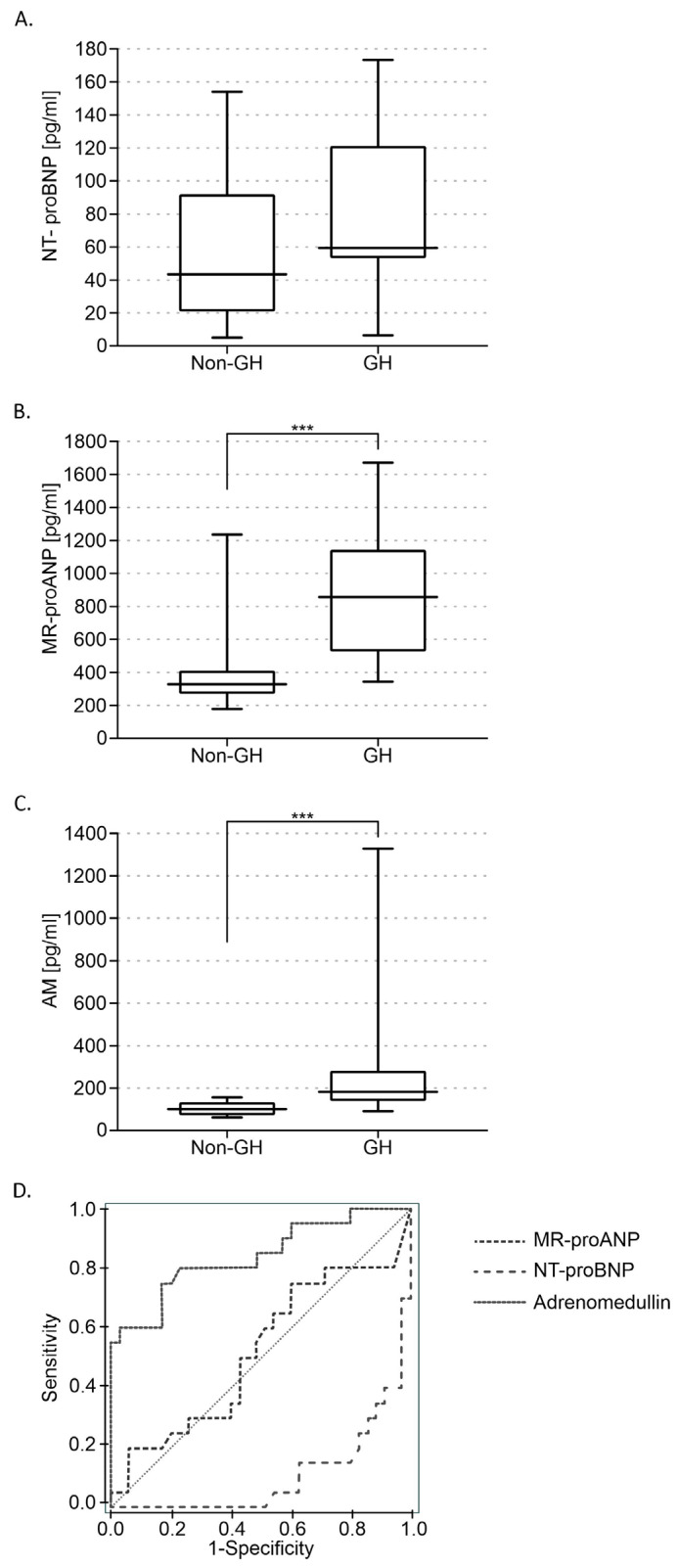
Levels of N-terminal pro-B-type natriuretic peptide (NT-proBNP) (**A**), midregional pro-atrial natriuretic peptide (MR-proANP) (**B**), and adrenomedullin (AM) (**C**) in patients without (non-GH) and with gestational hypertension (GH). Receiver operating characteristic (ROC) curves showing the predictive power of AM, MR-proANP, and NT-proBNP levels (**D**). The area under the curve (AUC) for AM was: 0.839 (95% CI 0.722–0.954, *p* < 0.001); for MR-proANP: 0.519 (95% CI 0.353–0.684, *p* < 0.001), and for NT-proBNP: 0.104 (95% CI 0.022–0.187, *p* = 0.82). Mean ± SE are shown. *** *p* < 0.001.

**Figure 2 diagnostics-14-02670-f002:**
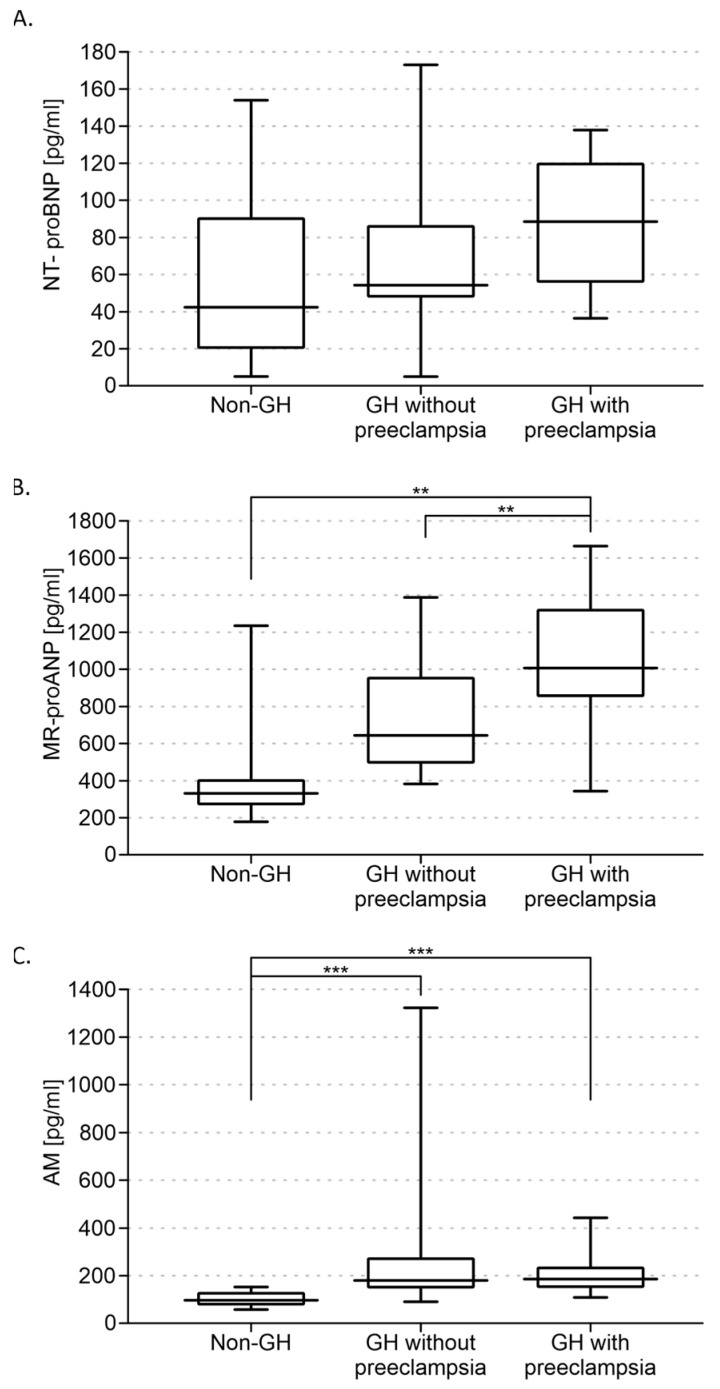
Levels of N-terminal pro-B-type natriuretic peptide (NT-proBNP) (**A**), midregional pro-atrial natriuretic peptide (MR-proANP) (**B**), and adrenomedullin (AM) (**C**) in patients without gestational hypertension (non-GH) and with gestational hypertension without pre-eclampsia (GH without pre-eclampsia) and with pre-eclampsia (GH with pre-eclampsia). Mean ± SE are shown. ** *p* < 0.01; *** *p* < 0.001.

**Table 1 diagnostics-14-02670-t001:** Comparison of the control group (non-GH) and the gestational hypertension (GH) group. BMI—Body mass index.

	Non-GH (*n* = 41)	GH (*n* = 18)	*p* Value
Age (years)	30	33	*p* < 0.001
Weight (kg)	65 ± 1.53	78.2 ± 3.5	*p* < 0.001
BMI (kg/m^2^)	22 ± 0.55	28 ± 1.34	*p* < 0.001
First birth (%)	42.4%	50%	-
Second birth (%)	48.5%	50%	-
Third birth (%)	9.1%	-	-
Pregnancy duration (weeks)	38.8 ± 0.49	36.6 ± 0.55	*p* < 0.0016
Natural childbirth (%)	68.5%	30%	*p* < 0.001
Cesarean section (%)	31.5%	70%	*p* < 0.001
Newborn’s body weight (g)	3389 ± 99	3016 ± 199	*p* < 0.045
Neonatal Apgar Score	9.97 ± 0.03	9.30 ± 0.24	*p* < 0.001

## Data Availability

The original contributions presented in the study are included in the article, further inquiries can be directed to the corresponding author/s.

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
