# Peer review of "High Serum Adrenomedullin and Mid-Regional Pro-Atrial Natriuretic Peptide Concentrations in Early Pregnancy Predict the Development of Gestational Hypertension"

_diagnostics, 2024, doi:10.3390/diagnostics14232670_

Round 1

Reviewer 1 Report

Comments and Suggestions for Authors

-Authors work “ High serum adrenomedullin and MR-proANP concentrations 3 in early pregnancy predict development of gestational hyper-4 tension

-I think in the statistical analysis section, logistics regression and corelation analysis should be done in the analysis and maybe  ROC analysis is good in such studies

- ELISA kits should be more detailed in Laboratory studies  forexample : The serum

 adrenomedullin concentrations were detected by the enzyme immunoassay method

(Sunredbio, Jufengyuan Road, Baoshan District, Shanghai China catalog number: 201-12-5569) and the intra-assay CV was <10% and the inter-assay CV was <12%.

-The work should be reviewed for grammatical and spelling errors

Author Response

REVIEW 1

The authors of the manuscript would like to thank the reviewers for their critical and valuable comments. Corrections have been made to the text. We hope that the reviewers will find the manuscript acceptable in its present form.

Comment 1: I think in the statistical analysis section, logistics regression and corelation analysis should be done in the analysis and maybe  ROC analysis is good in such studies

Authors: ROC curves and area under the curves (AUC) were plotted to assess the sensitivity and specificity of the measured parameters (Figure 1D). Our version of Statistica requires the purchase of an add-on to count logistic regressions. We will not be able to fit this into the timeframe of the review.  

Comment 2:  ELISA kits should be more detailed in Laboratory studies  for example : The serum

 adrenomedullin concentrations were detected by the enzyme immunoassay method (Sunredbio, Jufengyuan Road, Baoshan District, Shanghai China catalog number: 201-12-5569) and the intra-assay CV was <10% and the inter-assay CV was <12%.

Authors:  thank you for suggestion. Corrections has been made lines: 171-178

Comment 3: The work should be reviewed for grammatical and spelling errors

Authors: The text has been edited and proof-read by a native speaker.

Reviewer 2 Report

Comments and Suggestions for Authors

This work is interesting small stone from mosaic of preeclampsia study. IT has scientific sound. Number of participations looks as too small but statistic evaluation dont support my feeling.

I reccomend add to introduction description of today´s screening for preeclampsia risk in first trimestr and resasons for looking for another markers.

In summary is a plenty of abreviation.  Especially in case of ANP a BNP it is not clear, where is difference. It could be improve.

Author Response

REVIEW 2

COMMENT 1: This work is interesting small stone from mosaic of preeclampsia study. IT has scientific sound. Number of participations looks as too small but statistic evaluation dont support my feeling.

Authors: The authors of the manuscript would like to thank the reviewers for their critical and valuable comments. Corrections have been made to the text. We hope that the reviewers will find the manuscript acceptable in its present form. In the beginning, we had a fairly large group of patients, but this has been reduced due to other diseases or a change of centre. We plan to continue the study and expand it to analyze additional measurements.

COMMENT 2: I recommend add to introduction description of today´s screening for preeclampsia risk in first trimestr and resasons for looking for another markers

Authors:  A paragraph on current guidelines for the identification of risk factors for pre-eclampsia in the first trimester has been added to the introduction. (lines: 119-140)

COMMENT 3: In summary is a plenty of abreviation.  Especially in case of ANP a BNP it is not clear, where is difference. It could be improve.

Authors: To reduce the number of abbreviations (ANP/BNP) in discussion, the text has been amended in several places.

Round 2

Reviewer 1 Report

Comments and Suggestions for Authors

The figure may have been missing, I couldn't see the figure 1

Comments on the Quality of English Language

No comments

Author Response

Comment: The figure may have been missing, I couldn't see the figure 1

Response 1: Please accept my apologies. I have attached the Figure as a separate file. It is now inserted into the text. 
